# Inhibition of RPS6K reveals context-dependent Akt activity in luminal breast cancer cells

**Cemal Erdem** [1,2¤], **Adrian V. Lee** [3,4,5], **D. Lansing Taylor** [1,2], **Timothy R. Lezon** [1,2]*

**1** Department of Computational & Systems Biology, University of Pittsburgh, Pittsburgh, Pennsylvania, United States of America, **2** University of Pittsburgh Drug Discovery Institute (UPDDI), University of Pittsburgh, Pittsburgh, Pennsylvania, United States of America, **3** Department of Pharmacology & Chemical Biology, UPMC Hillman Cancer Center, University of Pittsburgh, Pittsburgh, Pennsylvania, United States of America, **4** Magee-Womens Research Institute, Pittsburgh, Pennsylvania, United States of America, **5** The Institute for Precision Medicine, Pittsburgh, Pennsylvania, United States of America

¤ Current address: Department of Chemical & Biomolecular Engineering, Clemson University, Clemson, South Carolina, United States of America
* lezon@pitt.edu

**Data Availability Statement:** All relevant data are within the manuscript and its Supporting Information files.

**Funding:** This work was funded by NIH grants P30 CA047904 and U01CA204826 (DLT), and

## Abstract

Aberrant signaling through insulin (Ins) and insulin-like growth factor I (IGF1) receptors contribute to the risk and advancement of many cancer types by activating cell survival cascades. Similarities between these pathways have thus far prevented the development of pharmacological interventions that specifically target either Ins or IGF1 signaling. To identify differences in early Ins and IGF1 signaling mechanisms, we developed a dual receptor (IGF1R & InsR) computational response model. The model suggested that ribosomal protein S6 kinase (RPS6K) plays a critical role in regulating MAPK and Akt activation levels in response to Ins and IGF1 stimulation. As predicted, perturbing RPS6K kinase activity led to an increased Akt activation with Ins stimulation compared to IGF1 stimulation. Being able to discern differential downstream signaling, we can explore improved anti-IGF1R cancer therapies by eliminating the emergence of compensation mechanisms without disrupting InsR signaling.

## Author summary

The activity of the type I insulin-like growth factor receptor (IGF1R) has been linked to aggressive cancer growth and spreading, and inhibiting IGF1R activity has slowed cancer growth in laboratory models. Unfortunately, molecules that are known to bind to IGF1R also bind to the closely related insulin receptor (InsR) and can disrupt metabolism in a fashion similar to diabetes. To alter the activity of IGF1R without affecting that of InsR, one must identify differences in how the two receptors affect other proteins. Here we build a computational model of the biochemical signaling pathways initiated by the two receptors. By fitting the model to data, we identify potential differences in IGF1R and InsR signaling—specifically, our model predicts that feedback from the ribosomal protein

1UL1TR001857 (TRL). The funders had no role in study design, data collection and analysis, decision to publish, or preparation of the manuscript.

**Competing interests:** The authors have declared that no competing interests exist.

S6 kinase (RPS6K) to the insulin receptor substrate (IRS) protein is sensitive to whether IGF1R or InsR is activated. We experimentally validate our model's predictions in three breast cancer cell lines, leading to possible new targets for IGF1R-associated cancer therapy.

## Introduction

Insulin and type I insulin-like growth factor (IGF1) are closely related hormones critical to development and metabolism [1–4]. Their receptors, InsR and IGF1R, are structurally and functionally similar, showing 60% overall amino acid sequence similarity and 84% identity at the kinase domain [5,6]. The signaling pathways of both receptors have been linked to cancer, where both can activate proliferation and survival cascades. Increased insulin and IGF1 levels have been shown to correlate with an increased risk of several cancer types [7–9]. IGF1R content in breast cancer (BRCA) tumors is 14 times higher than in normal tissue, and inhibiting IGF1R has been shown to block tumor growth in cell lines and model organisms [10–15].

While no recurrent cancer-specific mutations of IGF1R or its ligands have been described to date, studies have provided evidence for a link between this signaling pathway and the risk of developing cancer. IGF1R signaling leads to proliferative and anti-apoptotic signaling by employing Ras/MAPK and PI3K/Akt cascades. Clinical trials of IGF1R targeting received some positive responses; however, compensation mechanisms emerge and decrease the efficacy of such drugs [16–18]. Insulin stimulates cell growth, differentiation and promotes synthesis while inhibiting the lysis of macromolecules [4]. Insulin malfunctioning results in dysregulation of these processes and causes elevated glucose and lipid levels. Insulin resistance has been associated with type II diabetes and obesity, where increased insulin levels are shown to correlate with increased risk of several cancer types [4,7,8,19].

The insulin and IGF1 receptors are heterotetramers, or rather a dimer of heterodimers, with two α and two β subunits. Beta subunits contain the intracellular kinase domains. The α-subunits span the extracellular ligand-binding domains. IGF1R and InsR can also form hybrid receptors, with one α-β pair from each. These hybrid receptors show a differential affinity for the three ligands [20]. Two recent studies suggested that the extracellular domains (ECD) of the apo-receptor forms exert a physical force to keep the intracellular kinase domains apart from each other, enough to prevent auto-phosphorylation [21,22]. Ligand binding then induces a conformational change that lets the transmembrane and kinase domains interact and auto-phosphorylate. The changes upon ligand binding are studied by Houde and Demarest [23]. Kiselyov et al. used modeling approaches to re-capture available ligand binding dynamics of insulin [24]. This study only considered the ligand-receptor binding events, ignoring the fact that downstream elements of the transduced signal also affect available receptors on the cell surface.

Understanding the differences between the highly similar InsR and IGF1R signaling is essential for therapeutic development and clinical trial design. In a previous study [25], we constructed statistical models of insulin and IGF1 signaling from an extensive proteomics data set. Our models revealed, and experiments confirmed, cell-level differences in signaling through the two pathways. Specifically, acetyl-CoA carboxylase (ACC) or E-Cadherin knockdown increases MAPK or Akt phosphorylation, respectively, in IGF1 stimulated cells over Ins stimulated cells [25]. Subsequent work showed that loss of E-Cadherin increases the sensitivity of breast cancer cells to IGF1R targeted therapy by hyperactivating the IGF1R signaling pathway [26]; nevertheless, neither ACC nor E-cadherin is canonical members of these signaling

pathways and the precise mechanisms through which these proteins influence signaling remains hidden.

We construct a mechanistic model of signaling through the two receptors to identify known pathway components that directly influence differences between InsR and IGF1R signaling. Most existing models treated IGF1R and InsR signaling identically [27–30], while some, although they modeled them individually, did not focus on studying them [31]. In contrast, our model retains each receptor's unique identity and recovers differences in signaling through IGF1 and Ins. We construct our model and generate the simulation files using the BioNetGen rule-based modeling software [32]. Our training dataset includes phospho-proteomic data of IGF1 and Ins stimulated cells (see Methods for more information). We also use an independent test dataset (of drug response) to validate our model performance. After model validation, the subsequent systematic parameter scanning suggested that perturbing ribosomal protein S6 kinase (RPS6K) activity should increase Akt activation with insulin stimulation compared to IGF1 stimulation. Finally, we experimentally confirm this prediction. Our work demonstrates that modulating targets downstream of IGF1R and InsR may provide an alternative to specifically modulating IGF1R, potentially allowing targeted IGF1R therapies that do not disrupt insulin signaling and glucose metabolism.

## Results

### Computational model

We developed our model (Fig 1A) from previous work in which only IGF1R signaling [27,28] or InsR signaling [29,30] was modeled. Ligand binding to IGF1 or insulin receptors promotes receptor intracellular domains to auto-phosphorylate, leading to IRS and SOS binding and activation [27–29,33,34]. Phosphorylated IRS can also activate SOS in addition to PI3K [29,35,36]. SOS activation leads to activation/phosphorylation of Ras, Raf, MEK, and MAPK (ERK) [10,27]. PI3K activation causes PDK1, Akt, mTORC1, and RPS6K activation. Within

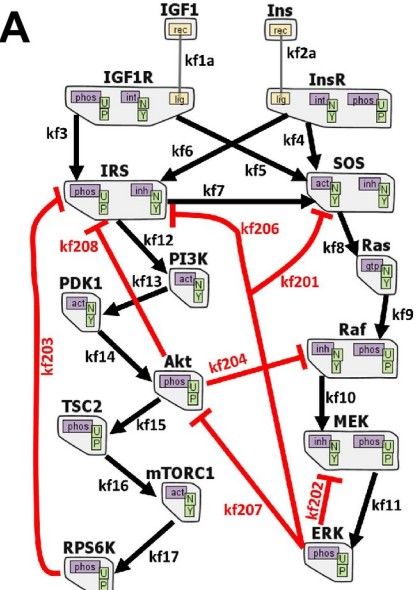

| No | Species | Description | Value Reference | Parameter Range (mpc) | MCF7 (best-fit) |
|----|---------|-------------|-----------------|-----------------------|-----------------|
| 1 | IGF1 | IGF1 | varied/set | 100000 | **100000** |
| 2 | Ins | Insulin | varied/set | 100000 | **100000** |
| 3 | IGF1R | Total IGF1R molecules | fit | 500-50000 | **25000** |
| 4 | InsR | Total InsR molecules | fit | 500-50000 | **25000** |
| 5 | IRS | Total IRS molecules | fit | 20000-240000 | **92766** |
| 6 | SOS | Total SOS molecules | fit | 20000-240000 | **90075** |
| 7 | Ras | Total Ras molecules | fit | 20000-240000 | **230642** |
| 8 | Raf | Total Raf molecules | fit | 20000-240000 | **126069** |
| 9 | MEK | Total MEK molecules | fit | 300000-1200000 | **1098164** |
| 10 | ERK | Total ERK molecules | fit | 300000-1200000 | **763172** |
| 11 | PI3K | Total PI3K molecules | fit | 20000-240000 | **64009** |
| 12 | PDK1 | Total PDK1 molecules | fit | 20000-240000 | **186081** |
| 13 | Akt | Total Akt molecules | fit | 300000-1200000 | **432907** |
| 14 | TSC2 | Total TSC2 molecules | fit | 20000-240000 | **131339** |
| 15 | mTORC1 | Total mTORC1 molecules | fit | 20000-240000 | **83469** |
| 16 | RPS6K | Total RPS6K1 molecules | fit | 20000-240000 | **121978** |

**Fig 1. The computational mechanistic model representation.** (A) The topology of the dual IGF1R/InsR signaling network is illustrated. The model includes 16 proteins. Black arrows represent activation, and red lines indicate inhibition of the corresponding active molecule. The graph is adapted from RuleBender software. (B) The initial molecule numbers of the species in the model. The ranges of parameter values and the "best-fit" values are reported. mpc: molecules per cell.

the cells, Akt phosphorylates and inactivates TSC2, where unphosphorylated TSC2 inactivates RHEB, an activator of mTORC1 [37,38]. Here, we modeled the mTORC1 activation by assuming the Akt phosphorylated TSC2 as the activator of mTORC1. There are numerous negative feedback loops and crosstalk within the system (Fig 1A) [28,39].

The constructed model has 14 proteins and two ligands, and 66 parameters, of which 16 are the total protein counts (Fig 1B and S1 Table). The protein counts, and 34 of the rate parameters, are common between IGF1 and insulin models, where each model has eight specific parameters. The parameter ranges are selected based on previous literature [40–43]. See BNGL model in S2 Text and corresponding state variables and corresponding Ordinary Differential Equations (ODEs) in S3 Text. The mechanistic model in this work was constructed using rule-based modeling with BioNetGen and RuleBender software [32,44–46].

## The computational training performance of the model

The experimental dataset used for model training contains proteomic (RPPA) data of phosphorylated and total protein levels at six different discrete time points (0–48 hours). Most of the activation (or phosphorylation) events in the downstream cascades occur within minutes post-treatment. By studying signaling in the earlier time points, we hope to unravel some *early response differences* of the breast cancer cells to single IGF1 or insulin stimulation.

The parameter estimation procedure using Markov chain Monte Carlo (MCMC) sampling and parallel tempering [47] yields ensembles of parameter sets and corresponding posterior distributions (S2 and S3 Figs). The resulting parameter set (S1 Table) of the network model was estimated, and simulation results conveyed qualitative and quantitative agreement to experimental data for both IGF1R and InsR signaling (Fig 2A). The dashed lines represent the performance of the best-fit parameter set model on training data. In addition to the best-fit models of IGF1 and insulin stimulation, the ODE models were simulated for an ensemble of parameter sets (area plots in Fig 2A). This approach corresponds to screening a virtual population of cells, where each parameter set has different initial conditions set for the simulation. The computational model recaptures experimental data within the specified range of parameters, indicating a decent set of rate parameters.

Additionally, a sensitivity analysis of the model was carried out, and details are described in S1 Text. The cascade-specific parameters affected the corresponding readout the most, and the parameters of receptor kinetics induced changes in receptor phosphorylation levels (S4–S7 Figs).

There are other mechanistic models in the literature [28,31,48], spanning different proteins and interactions (edges) among the protein species in our model. We specifically tested individually adding the interactions depicted in [28] into our model. Parameter scanning these model variants did not yield an improvement in data fitting, and thus the candidate interactions are discarded from the final model topology.

## IGF1 dose-response and PI3K inhibition in MCF7 cells

We tested the model performance using two independent datasets: IGF1 dose-response (Fig 2B and S2 Table) and PI3K/mTORC1 inhibition (Fig 2C and S3 Table). The simulation results for IGF1 dose-response were obtained by only changing the level of IGF1 input into the system. The results were within the range of experimental error and showed qualitatively good agreement with the data, although there were some differences between model predictions and experimental data, especially in MAPK response to low doses of ligands. We attribute these discrepancies to simplifying assumptions of the MAPK signaling, such as omitting scaffold proteins and double-phosphorylation events. However, our model recapitulated the system dynamics at the higher IGF1 concentrations. Our training data were collected at 10 nM IGF1

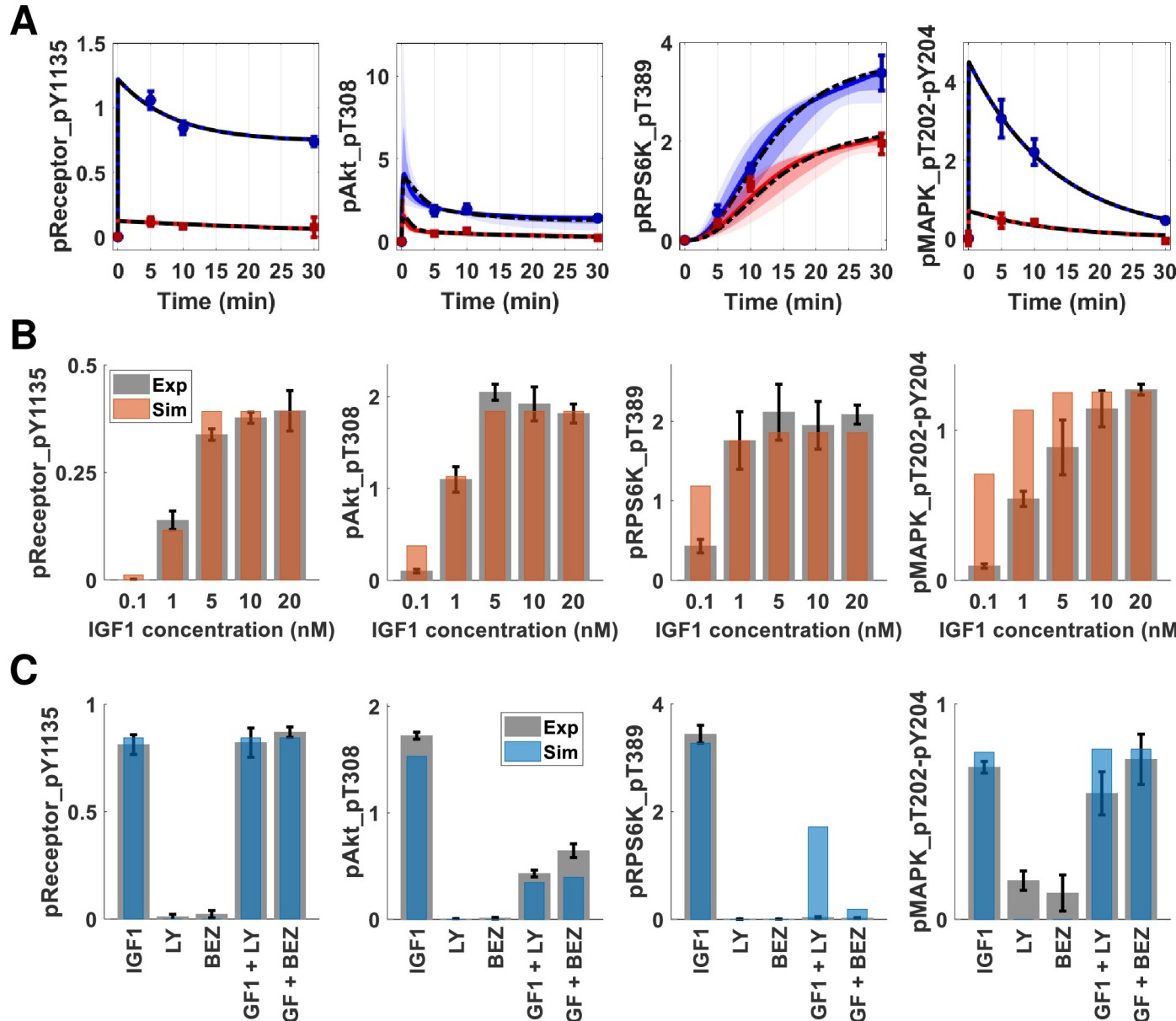

**Fig 2. The mechanistic model training and test performance on multiple omics datasets show qualitative and quantitative agreement.** (A) The time course trajectories of IGF1 (blue) and insulin (red) stimulations. The ODE model was simulated 10000 times with an ensemble of parameter sets with different total protein numbers. The initial amounts of proteins were sampled using Latin Hypercube Sampling. The plots show 5%-95% (light) and 15%-85% (dark) confidence intervals for IGF1 (blue) and insulin (red) models. The circles with error bars are the corresponding RPPA data points. The dashed black lines are the trajectories for "best-fit" parameter sets. (B) The computational model (orange bars) recapitulates experimental IGF1 dose-response data (gray bars). The error bars represent the standard error of the mean from three independent biological replicates. The y-axis represents scaled protein numbers. (C) The computational model (blue bars) recapitulates PI3K inhibition data (gray bars). The columns of the x-axis correspond to IGF1: IGF1 (10 nM) stimulation, LY: first inhibitor only, BEZ: second inhibitor only, IGF1 +LY: first inhibitor and IGF1 (10 nM), and IGF1+BEZ: second inhibitor with IGF1 (10 nM) stimulation. The receptor and MAPK phosphorylation are not affected by PI3K or mTORC1 inhibition, whereas Akt and S6 kinase phosphorylation decrease. The inhibitions are simulated in the computational models as a 90% reduction in the corresponding rate constant(s). The error bars represent the standard error of the mean from three independent replicates. The y-axis represents scaled protein numbers.

stimulation, a concentration at which the simulations and data agreed within experimental replicate error. The model overall thus provided satisfactory performance on IGF1 dose response, capturing our observation via a single parameter (IGF1 dose input) variation.

The second tier of the performance test was done using PI3K and mTORC1 inhibition data. One specific PI3K inhibitor, LY294002 (LY), and one dual inhibitor of PI3K and mTORC1, BEZ235 (BEZ), were administered alone or in combination with IGF1 (10 nM) (S3 Table). To simulate the activity of the PI3K inhibitor LY294002, we decreased the rate parameter value controlling PI3K-mediated activation of PDK1 by 90%. To simulate the activity of the PI3K/mTORC1 inhibitor BEZ235, this rate constant and the constant that controls S6K activation through mTORC1 were both decreased by 90%. Neither inhibitor affected any downstream signaling (Fig 2C gray bars-experimental data, blue bars-simulations). The addition of IGF1 into the inhibitor-treated system produced decreased phosphorylation of Akt compared to treatment with IGF1 alone. Although ribosomal protein S6 kinase phosphorylation was diminished in both inhibitor experiments, the computational model predicted a non-zero activation for IGF1 stimulated cells with PI3K inhibitor LY. Even with 90% inhibition of PI3K, minimal mTORC1 activation occurred and led to phosphorylation of S6K. The experimental observation might result from the fact that the kinase inhibitors are dirtier than the simple perturbations simulated here [49]. The model recapitulated complete deregulation of S6K phosphorylation in the case of another inhibitor (BEZ) with multiple targets. Overall, the model showed qualitative agreement with the experimental results and captured the expected behavior of the perturbed systems, with minimal parameter perturbations.

## Modeling suggests experiments to disentangle InsR signaling and IGF1R signaling

We performed simulations in which kinetic parameters were individually perturbed over several orders of magnitude. Comparing the MAPK and Akt responses between the perturbed and unperturbed models for insulin and IGF1 stimulations (Fig 3A), we identified three interactions that may distinguish InsR signaling from IGF1R signaling:

1. The model predicts that increasing the rate of SOS activation by IRS protein (rate parameter k7) will increase MAPK phosphorylation more in IGF1-treated cells than in Ins-treated cells (Fig 3A top panel).

2. Turning off rate parameter kf208, corresponding to the feedback of Akt on IRS, caused differential up-regulation of Akt activation (Fig 3A middle panel).

3. The model predicted that upon inhibiting S6 kinase, the insulin-stimulated cells would have increased Akt phosphorylation at 30 min. The increase would be larger than that in the IGF1 stimulated cells (Fig 3A bottom panel). The rate parameter "kf203" controls negative feedback from S6 kinase on IRS.

All three computational predictions pointed out that IRS regulation is critical for differential downstream regulation of IGF1 and insulin receptor cascades (Fig 3B).

**Ribosomal protein S6 kinase (RPS6K) inhibition in luminal BRCA cells.** We experimentally validated the last prediction by chemically inhibiting ribosomal protein S6 kinase in breast cancer cells and then treating them with IGF1 and insulin, as described in *Methods*. The experiments were performed in MCF7, T47D, and ZR75-1 cell lines, all of which are luminal and hormone receptor positive subtype. The decrease in ribosomal protein S6 phosphorylation (by RPS6K) levels was most remarkable in MCF7 cells (Fig 4). However, in all three cell lines, the level of Akt phosphorylation (S473) increased in insulin-stimulated drug-treated cells, compared to IGF1 stimulated cells. This result follows the computational prediction of the mechanistic model.

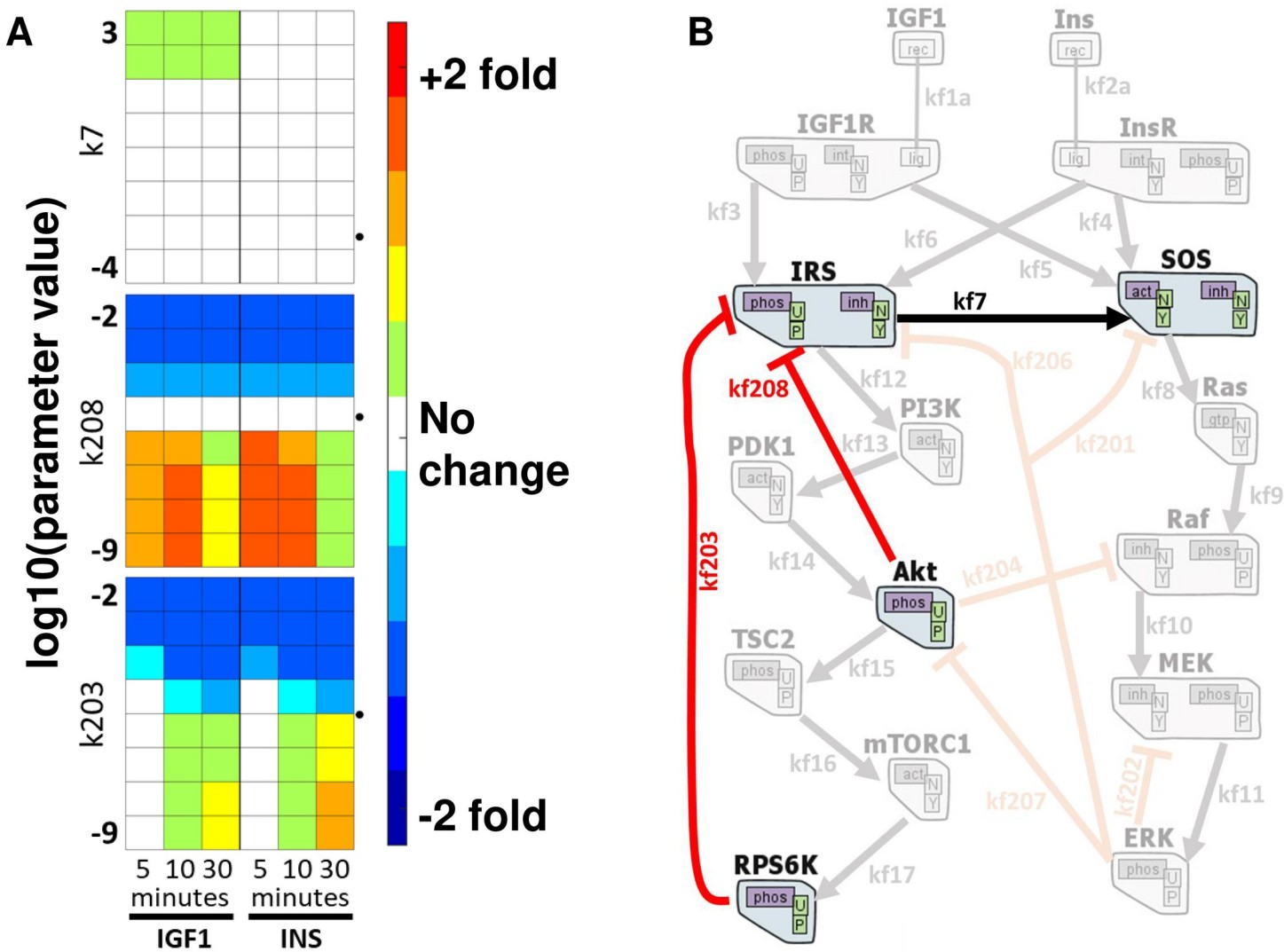

**Fig 3. *In silico* perturbation of rate constants reveal differential effects on Akt and MAPK phosphorylation.** The model output of pMAPK and pAkt levels are determined by setting and changing the value of each parameter individually, from zero to infinity. Each box above is for a parameter, and each column is for a single time point response of 5, 10, and 30min. The left three columns are from IGF1 stimulation results, and the right three columns are from insulin stimulated simulations. The rows represent the log10 of the set parameter value, and the colors denote the log2 fold change from the unperturbed model output (red: up-regulation, blue: down-regulation). The dotted arrowheads indicate the unperturbed model parameter values in log10. The rest of the parameter perturbation scanning results are shown in S8 Fig. (B) The respective edges from (A) are depicted together to emphasize that all perturbations leading to differential regulation are concentrated on IRS proteins.

## Discussion

Understanding differences between InsR and IGF1R signaling may aid the development of cancer therapeutics. Our earlier work showed that acetyl-CoA carboxylase knock-down increases MAPK phosphorylation while E-cadherin knock-down promotes higher Akt activation in IGF1 stimulated cells [25]. These were novel findings that show how insulin and IGF1 downstream signaling cascades differ in cancer cells; however, the mechanisms for these differences were obfuscated by the distance of E-cadherin and ACC from the InsR and IGF1R signaling pathways. In this work, we used our newly constructed mechanistic model to identify pathway-linked mechanisms that may influence distinct responses upon IGF1 and insulin administration. The model predicts that feedback from ribosomal protein S6 kinase (p70S6K) on IRS differentially affects Akt activation under IGF1 and insulin-stimulated cells.

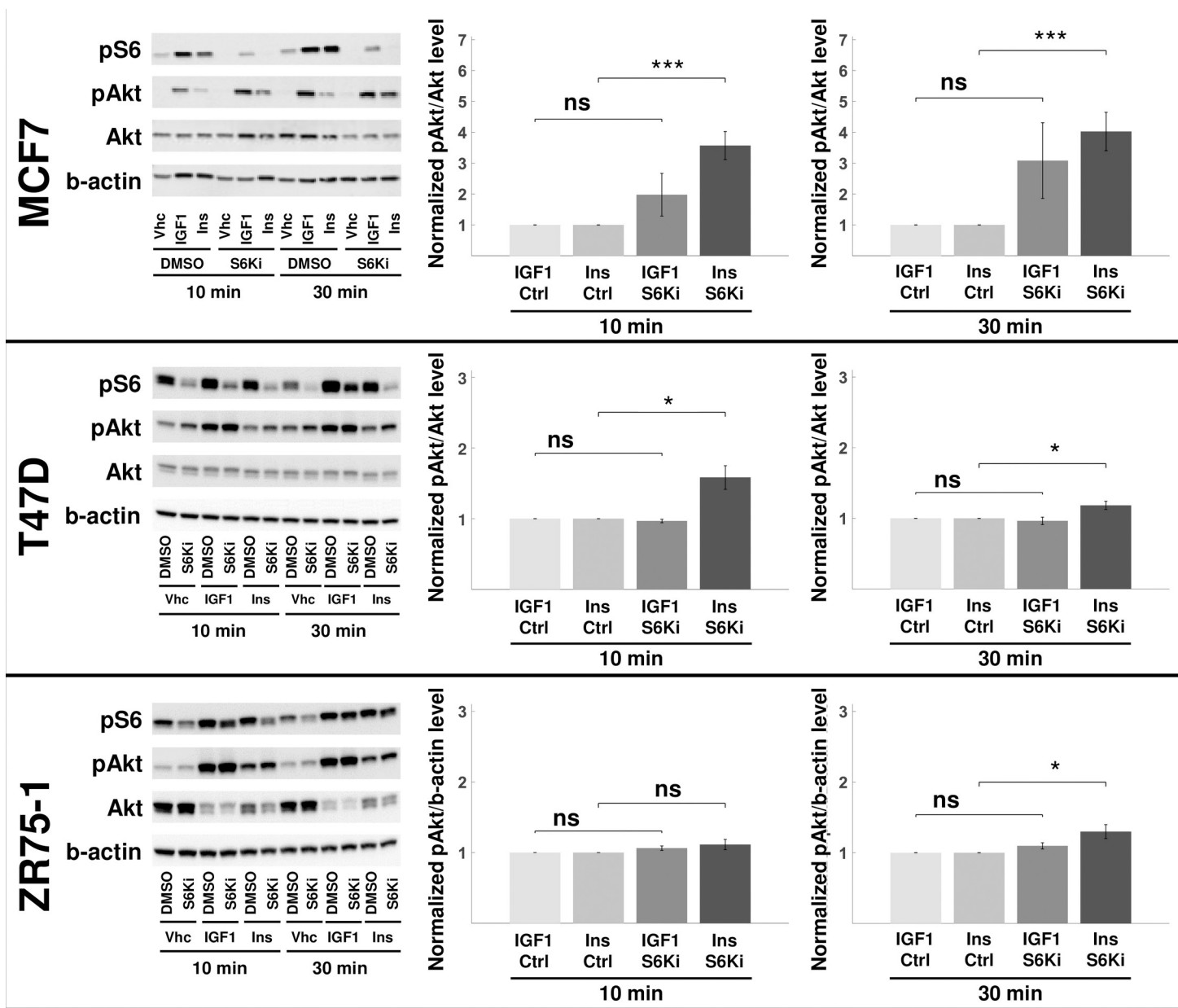

**Fig 4. Ribosomal S6 kinase inhibition up-regulates Akt phosphorylation.** The RPS6K was inhibited in MCF7, T47D, and ZR75-1 cells. The pS6 levels are used as the proxy for S6K inhibition efficiency. The total Akt and b-actin levels are used to normalize pAkt and pS6 levels, respectively. The perturbagen, ligand, and time point for each sample are listed below the blot images (the rest of the replicates are shown in S9 Fig). The blot quantifications are shown on the right. The values are reported as normalized to the corresponding no-inhibition control (ctrl). pAkt levels represent the response of the cells to the perturbation. All MCF7, T47D, and ZR75-1 cell lines showed higher up-regulation of pAkt in insulin-stimulated cells at 30 min. The results are compared using an unpaired, one-tailed two-sample t-test, and P<0.05 (*), P<0.01 (**), P<0.005 (***), nonsignificant (ns). Results shown are mean ± s.e.m. of four independent replicates. S6 phosphorylation quantifications at 10 and 30 min are shown in S10 Fig.

Inhibition of RPS6K activity in three cell lines induced a difference in the regulation of Akt activity in response to different stimuli, with a more significant change induced by insulin rather than by IGF1. This experimental work approximates the precision tuning of a single feedback interaction in the mathematical model. Although the model predicts that IRS plays a role in the observed behavior, we acknowledge that the exact mechanism through which RPS6K inhibition induces differential signaling remains unknown. A substrate other than IRS

may be responsible for the effect; however, IRS is central to other model predictions, and the literature supports its role. IRS is an adaptor protein that is considered one of the bottlenecks of signaling activation [50,51], and recent structural analysis of the two RTKs suggests a differential binding of IRS proteins [34]. Relieving the negative feedback from S6K on IRS was previously shown to sensitize colorectal cancer cells to EGFR inhibition [28].

One important observation from our experiments is that pAKT activation in response to IGF1 was much stronger than to Ins in MCF7 cells. As a consequence, a mechanism of reaching maximal phosphorylation upon IGF1 treatment (with or without S6K inhibition) was possible and could have clouded our observations at 30min. To circumvent such an issue, we selected two additional cell lines with pAKT response to both IGF1 and Ins [25] to repeat our validation experiments. Importantly, the phosphorylation profiles in ZR75-1 cells with 10 nM IGF1 and Ins are very close each other (S11 Fig). Since the inhibition of p70S6K in these cells also conveyed a more significant pAKT increase in insulin-stimulated cells, we can conclude that our results hold true for a variety of cell lines and basal pAKT response profiles.

We selected a level of detail in our model that would allow us to generate testable hypotheses. Our model included downstream MAPK and Akt cascades, with negative feedback loops. Our model assumed that the reactions are linear and either binary interactions or simple degradation/inactivation/activation events. Both ligand binding events were reversible. The internalized ligands (bound to the receptor) were degraded when the receptors were recycled back to the cell surface. We represented ERK and Akt activation events as single phosphorylation events and considered a simplified version of the mTORC1 activation by TSC2. For the negative feedback events, among the possible mechanisms reported in the literature [27,28], we only included the interactions yielding improved data fitting. We trained our model on proteomic signaling data in MCF7 breast cancer cells and tested it against data from two different drug inhibitions. We then used our model to interrogate differential IGF1 and insulin stimulation responses. In the end, the model served its purpose by suggesting experiments that yielded novel results.

We studied how IGF1 and Ins activate their cognate receptors. Indeed, the receptors have different isoforms, IGF1R / IGF2R and InsR-A / InsR-B [20]. IGF1R binds to both IGF1 and IGF2 (insulin-like growth factor 2). The latter ligand is mostly fetal and not studied in this work. IGF2R, the IGF1R homolog, has no kinase domain and is considered to sequester IGF2 primarily [20]. One isoform of insulin receptors, the InsR-A, is functional in fetal tissues and cancer cells, whereas InsR-B isoform is expressed in adults. InsR-A has a similar affinity for IGF2 and insulin [6,11]. There are even *heterodimers* of IGF1R/InsR [20], which can ideally bind any of the ligands. In this work, we only focused on revealing how IGF1 and Ins stimulations exert different responses in breast cancer cells and not focused on the isoform specificity.

We believe that by determining ways to suppress IGF1R signaling alone, we can find more effective ways of preventing cancer and inhibiting its progression. Additionally, by excluding possible adverse effects of global inhibition of InsR signaling and combining IGF1R inhibition with existing targeted therapies, we might overcome the emergence of compensation mechanisms. Such efforts will be more important for cases like triple-negative breast cancer (about 10–15% of all breast cancers), which has limited treatment options and a worse prognosis.

The study of systems biology encompasses the employment of tools and techniques to extract information from large datasets. Within the quantitative systems pharmacologic (QSP) framework undertaken here, the mechanistic computational model is supplied with experimental and observational data and is iteratively refined [52–54]. The role of the models was then to simulate different stimulation conditions in silico and analyze the response with any possible off-target effects. In doing so, we can start stratifying patients to suitable personalized medicine treatments after recognizing and distinguishing that the IGF1R and InsR systems have different dynamics and novel signaling components.

## Methods

### Initial dataset and computational model parameter estimation

The initial dataset utilized here was defined in our previous work [25]. In short, reverse phase protein array (RPPA) is a high-throughput technique for quantifying levels of total and phosphorylated proteins [55,56]. The model parameters were estimated using RPPA expression levels of four phospho-proteins: pReceptor (both IGF1R and InsR), pAkt, pRPS6K, and pMAPK. Data from three early time points (5, 10, and 30 min) were used to estimate the parameters and calculate the fitting error.

In this work, all ODEs corresponding to the state variables were from BioNetGen [32], and we ran the simulations and analyzed the results in MATLAB (The MathWorks, Inc., version R2015a). We did the parameter estimation using Markov chain Monte Carlo (MCMC) sampling at different temperatures to search the parameter space both locally and globally [47]. High-temperature chains scan the parameter space more globally, and the probability of accepting an unfavorable move depends on the temperature. The swaps among different chains help avoid getting stuck in local minima. The approach samples the Bayesian posterior distribution of each parameter with uniform priors [57,58]. The estimation procedure outputs parameter ensembles for each chain. The minimum fitting error parameter set was defined as the "best-fit" and was used for all subsequent analyses (S1 Fig). See S1 Text for more details on the parameter fitting protocol.

### *In-silico* ensemble of cells

Using the value-ranges set for the initial protein count parameters, we generated an ensemble of 10,000 parameter sets using Latin hypercube sampling. By only changing the values of total protein numbers and keeping estimated rate parameters constant, different cell conditions were captured (e.g., a virtual cell population).

### Parameter perturbation scanning

Once the parameters were determined, simulations were run to analyze the response of the system. Each parameter was perturbed individually, and for every different value of each parameter, one simulation was run. The predicted levels of pMAPK and pAkt in the perturbed system were compared to the levels in the un-perturbed model output, simulating experimental knock-down (up-regulation) of proteins or reaction rates. Based on the results of the simulations here, perturbations that resulted in differential responses under IGF1 and insulin stimulation conditions were selected for further experimental exploration.

### Cell culture and immunoblotting

MCF7 (ATCC) cells were cultured in DMEM (ThermoFisher) with 10% FBS, plated on six-well plates at 400000 cells/well density. The cells, rested overnight, were serum-starved for 16–24 hours. Then, the cells were treated with DMSO control or ribosomal protein S6 kinase inhibitor LY2584702 (500 nM, Selleckchem) for three hours. Next, the cells were stimulated with control, IGF1 (10 nM), or insulin (10 nM) for 10 and 30 min. The cells were harvested, and protein concentration was quantified by BCA. Samples were collected using RIPA buffer (50mM Tris pH 7.4, 150mM NaCl, 1mM EDTA, 0.5% Nonidet P-40, 0.5% NaDeoxycholate, 0.1% SDS) with 1x HALT protease & phosphatase inhibitor cocktail (ThermoFisher). The immunoblotting was done using 12% acrylamide gels and PVDF membrane transfer (Millipore #IPFL00010, 0.45μm). Membranes were blocked in Odyssey PBS Blocking Buffer LI-COR and incubated in primary antibodies overnight: Akt S473 (Cell Signaling #4060; 1:1000),

total Akt (Cell Signaling #2920; 1:1000), phospho-S6 S235/236 (Cell Signaling #4858, 1:1000), total S6 (Cell Signaling #2217, 1:1000), and β-actin (Sigma #A5441; 1:5000). Membranes were incubated in LI-COR secondary antibodies for 1 hour (anti-rabbit 800CW, LI-COR #926–32211 or anti-mouse 680LT, LI-COR #925–68020; 1:10000) at room temperature. The imaging was done at LI-COR Odyssey Infrared Imager, where blots were quantified using LI-COR Image Studio Lite v5.2 software.

T47D (ATCC) and ZR75-1 (ATCC) cells were cultured in RPMI-1640 (HyClone, GE) with 10% FBS, 1% glutamine, and 1% Penicillin-Streptomycin, plated on six-well plates at 500000 cells/well density. The cells, rested overnight, were serum-starved for 16–24 hours. Then, the cells were treated with DMSO control or ribosomal protein S6 kinase inhibitor LY2584702 (1 μM) overnight. Next, the cells were stimulated with control, IGF1 (10 nM), or insulin (10 nM) for 10 and 30 min. The cells were harvested, and protein concentration was quantified by Bradford absorbance assay. Samples were collected using HEPES buffer (1% Triton X-100, 10% Glycerol, 5mM MgCl2, 25mM NaF, 1mM EGTA, 10mM NaCl) with 1x HALT protease & phosphatase inhibitor cocktail (ThermoFisher). The immunoblotting was done using 12% acrylamide gels (ThermoFisher #XP00125BOX) and PVDF membrane transfer (ThermoFisher #LC2002, 0.2 μm). Membranes were blocked in 5% milk in 1X TBST solution (TBST: Tris Buffered Saline (Sigma # T6664) with 0.1% Tween20) and incubated in primary antibodies overnight: Akt S473 (Cell Signaling #4060; 1:1000), total Akt (Cell Signaling #2920; 1:1000), phospho-S6 S235/236 (Cell Signaling #4858, 1:1000), total S6 (Cell Signaling #2217, 1:1000), and β-actin (Sigma #A5441; 1:5000). Membranes were incubated in HRP secondary antibodies for 45 min at room temperature (anti-rabbit, Jackson #111-035-003 or anti-mouse, Jackson #115-035-003; 1:8000). The imaging was done on a Philipps L4000 Imager using ECL substrates (BioRad #170–5060). The blots were quantified using LI-COR Image Studio Lite v5.2 software.

All three cell lines are parental cells obtained from ATCC, and internally mycoplasma tested regularly or when suspected of any contamination. No such observation during this study.

## Supporting information

**S1 Fig. Estimation error over four cycles PTEMPEST parameter fitting.** The trend shows a decrease of fitting error over the selected parameters. However, there are cases of acceptance for slightly worse performing parameter values and a very poor performing parameter value set is chosen once. Each dot above is a simulation error of a single parameter set, where only the values of the mentioned list of parameters are varied from the starting condition. (TIF)

**S2 Fig. The posterior parameter distributions for MCF7 mechanistic computational model.** Each subpanel histogram depicts posterior parameter distribution for one parameter in the model. Top 12 panels are an example on total protein levels, showing distributions from ~6000 parameter sets. The histograms for rate parameter values (from ~50000 parameter sets) are shown in log10 values in x-axes. Continued on S3 Fig. (TIF)

**S3 Fig. The posterior parameter distributions for MCF7 mechanistic computational model.** Each subpanel histogram depicts posterior parameter distribution for one parameter in the model. The histogram values (from ~50000 parameter sets) are shown in log10 values in x-axes. (TIF)

**S4 Fig. The parameter sensitivity analyses of Akt and MAPK responses.** Changing the individual parameter values up or down 2X causes changes in Akt and MAPK phosphorylation

levels. The values reported indicate the ratio of perturbed model response protein level to the level of the protein of un-perturbed model at 5min. Dark and light blue colors represent doubled parameter rate conditions with IGF1 and insulin stimulations, respectively. Orange and yellow colors represent halved parameter rate conditions with IGF1 and insulin stimulations, respectively. Bars are shown as concatenated for IGF1 and insulin conditions, where applicable. The two opposite parameter manipulations result in opposite changes in the phosphorylation levels, with very similar changes imposed in IGF1 and insulin.
(TIF)

**S5 Fig. The parameter sensitivity analyses of pRPS6K and pReceptor responses.** Changing the individual parameter values up or down 2X causes changes in S6 kinase and receptor phosphorylation levels. The values reported indicate the ratio of perturbed model response protein level to the level of the protein of un-perturbed model at 5min. Dark and light blue colors represent doubled parameter rate conditions with IGF1 and insulin stimulations, respectively. Orange and yellow colors represent halved parameter rate conditions with IGF1 and insulin stimulations, respectively. Bars are shown as concatenated for IGF1 and insulin conditions, where applicable. The two opposite parameter manipulations result in opposite changes in the phosphorylation levels, with very similar changes imposed in IGF1 and insulin.
(TIF)

**S6 Fig. The parameter sensitivity analyses of Akt and MAPK responses.** Changing the individual parameter values up or down 2X causes changes in Akt and MAPK phosphorylation levels. The values reported indicate the ratio of the AUC of the corresponding protein in perturbed model to the AUC of that protein in un-perturbed model, from time point zero to 30 min of stimulation. Dark and light blue colors represent doubled parameter rate conditions with IGF1 and insulin stimulations, respectively. Orange and yellow colors represent halved parameter rate conditions with IGF1 and insulin stimulations, respectively. Bars are shown as concatenated for IGF1 and insulin conditions, where applicable. The two opposite parameter manipulations result in opposite changes in the phosphorylation levels, with very similar changes imposed in IGF1 and insulin.
(TIF)

**S7 Fig. The parameter sensitivity analyses of pRPS6K and pReceptor responses.** Changing the individual parameter values up or down 2X causes changes S6 kinase and receptor phosphorylation levels. The values reported indicate the ratio of the AUC of the corresponding protein in perturbed model to the AUC of that protein in un-perturbed model, from time point zero to 30 min of stimulation. Dark and light blue colors represent doubled parameter rate conditions with IGF1 and insulin stimulations, respectively. Orange and yellow colors represent halved parameter rate conditions with IGF1 and insulin stimulations, respectively. Bars are shown as concatenated for IGF1 and insulin conditions, where applicable. The two opposite parameter manipulations result in opposite changes in the phosphorylation levels, with very similar changes imposed in IGF1 and insulin.
(TIF)

**S8 Fig. In silico perturbation of rate constants reveal differential effects on Akt and MAPK phosphorylation.** The ODE model output of pMAPK (A) and pAkt (B) levels are determined by setting and changing the value of each parameter individually, from zero to infinity. Each box above is for a parameter, and each column is for one time point response of 5, 10, and 30min. Left three columns are from IGF1 stimulation results and the right three columns are from insulin stimulated simulations. Rows represent the log10 of set parameter value. The colors represent the log2 fold change from un-perturbed model output. Red and blue respectively

indicate up and down regulation of the specified phosphorylation. The dotted arrow heads indicate the unperturbed model parameter values in log2.
(TIF)

**S9 Fig. Ribosomal S6 kinase inhibition up-regulates Akt phosphorylation.** The RPS6K is inhibited in MCF7 (top row), T47D (middle row), and ZR75-1 (bottom row) cell lines. The pS6 levels are used as the proxy for S6K inhibition efficiency in un-stimulated cells. The total Akt and b-actin levels is used to normalize pAkt and pS6 levels, respectively. The perturbagen, ligand, and time point for each sample are listed below the blot images.
(TIF)

**S10 Fig. Quantification of ribosomal S6 kinase inhibition.** The RPS6K is inhibited in three cell lines of MCF7 (top row), T47D (middle row), and ZR75-1 (bottom row). The values are reported as normalized to the corresponding no-inhibition control. S6 phosphorylation quantification at 10 and 30 min show inhibition efficiency The results are compared using unpaired, one-tailed two-sample t-test, and $P<0.05$ (*), $P<0.01$ (**), $P<0.005$ (***), nonsignificant (ns). Results shown are mean ± s.e.m. of four independent replicates.
(TIF)

**S11 Fig. Basal pAKT profiles of the three BRCA cell lines.** Time-course (phospho)-protein expression profiles. The change in the (phospho)-protein levels upon IGF1 or insulin stimulation, represents the difference in log2 expression levels (stimulated vs serum-starved). Six columns are shown for each cell line, each column representing one time point (5, 10, 30 min, 6, 24, 48 h). Red: up-regulation, blue: down-regulation. Adapted from [25].
(TIF)

**S1 Table. Model parameters.**
(XLSX)

**S2 Table. RPPA data for model construction and calibration.**
(XLSX)

**S3 Table. RPPA data for PI3K inhibitors.**
(XLSX)

**S4 Table. PTEMPEST parameter estimation configurations.**
(DOCX)

**S5 Table. Initial parameter values of the model.**
(DOCX)

**S1 Text. Parameter estimation details and supporting figures.**
(DOCX)

**S2 Text. BioNetGen model.**
(TXT)

**S3 Text. Model equations.**
(PDF)

## Acknowledgments

CE gratefully acknowledges Dr. Alison Nagle & Beth Knapick from Lee lab, and Laura Vollmer & Celeste Reese of UPDDI for wet-lab training.

## Author Contributions

**Conceptualization:** Cemal Erdem, Timothy R. Lezon.

**Data curation:** Cemal Erdem, Adrian V. Lee.

**Formal analysis:** Cemal Erdem, Adrian V. Lee, D. Lansing Taylor, Timothy R. Lezon.

**Funding acquisition:** Adrian V. Lee, D. Lansing Taylor, Timothy R. Lezon.

**Investigation:** Cemal Erdem, Adrian V. Lee, D. Lansing Taylor, Timothy R. Lezon.

**Methodology:** Cemal Erdem, Adrian V. Lee, Timothy R. Lezon.

**Project administration:** Adrian V. Lee, D. Lansing Taylor, Timothy R. Lezon.

**Resources:** Adrian V. Lee, D. Lansing Taylor, Timothy R. Lezon.

**Software:** Cemal Erdem, Timothy R. Lezon.

**Supervision:** Adrian V. Lee, D. Lansing Taylor, Timothy R. Lezon.

**Validation:** Cemal Erdem, Adrian V. Lee.

**Visualization:** Cemal Erdem.

**Writing – original draft:** Cemal Erdem, Adrian V. Lee, D. Lansing Taylor, Timothy R. Lezon.

**Writing – review & editing:** Cemal Erdem, Adrian V. Lee, D. Lansing Taylor, Timothy R. Lezon.

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
