## [Decision Letter · Decision Letter 0]

13 Apr 2021

Dear Dr. Lezon,

Thank you very much for submitting your manuscript "Inhibition of RPS6K reveals context-dependent Akt activity in luminal breast cancer cells" for consideration at PLOS Computational Biology.

As with all papers reviewed by the journal, your manuscript was reviewed by members of the editorial board and by several independent reviewers. In light of the reviews (below this email), we would like to invite the resubmission of a significantly-revised version that takes into account the reviewers' comments.

I would draw your attention specifically to the request of reviewer 2 to repeat a small subset of experiments to enable a better comparison between different effects.

We cannot make any decision about publication until we have seen the revised manuscript and your response to the reviewers' comments. Your revised manuscript is also likely to be sent to reviewers for further evaluation.

Sincerely,

Benjamin Hall

Guest Editor

PLOS Computational Biology

Florian Markowetz

Deputy Editor

PLOS Computational Biology

Please see the comments by the reviewers. I would draw your attention specifically to the request of reviewer 2 to repeat a small subset of experiments to enable a better comparison between different effects.

Reviewer's Responses to Questions

**Comments to the Authors:**

Reviewer #1: Erdem et al. describe a nice use of quantitative, continuous modelling to dissect differential response to insulin and IGF1 signalling. The authors predict that RPS6K is key to this differential response and test the mechanism to some degree experimentally. Due to the identical network links from InsR and IGF1R, these results would not be apparent from qualitative, discrete modelling of the system or without incorporating feedback into the network. I believe these results are of interest to breast cancer research community.

My comments relate to the presentation of the paper.

1. For much of the paper the exact modelling methodology used is unclear. The paper would be improved if there was a clear description of the BioNetGen/RuleBender/ODE based modelling approach used and an overview of model training against proteomics data upfront. The input, output, training and semantics of the model are not made precise as early as they could be. Some of this detail is already in the discussion and methods sections.

2. The discussion would benefit from comments on the implications of these results for breast cancer therapy.

Reviewer #2: In their manuscript “Inhibition of RPS6K reveals context-dependent Akt activity in luminal breast cancer cells”, Erdem et al. present a computational approach to find differences in the downstream signalling kinetics of the IGF and insulin signalling pathways. These heavily overlapping pathways are both implicated in cancer, and the authors argue that a better understanding of the kinetics of these pathways, might help to counteract compensatory activations following targeted therapy. As a proof of concept, they model the response of cells to IGF1 stimulation and the inhibition of downstream pathways. Through sequential in silico perturbations of the kinetic parameters governing the modelled reactome, the authors predict a more robust increase in AKT signalling 30 minutes after Ins stimulation compared to IGF1 stimulation, both under condition of S6K inhibition. This prediction is validated in cell lines.

Overall, I believe that the approach is sound and that the modelling is appropriate. In the following, I will summarise a few technical or conceptual difficulties that I see.

1. Accuracy of the model

The authors base their study on a number of previous studies and modelling efforts, and for this particular effort concentrate their mechanistic model on a few pathways that are downstream of both the IGF1R and the InsR. The main signalling axes encompasses the Akt-mTor axis, as well as the Ras-Raf-Mapk pathways, including several feedback-loops between them. This might seem a reductionist approach, but to me it appears an appropriate approach for a targeted question.

This assertion is largely vindicated in the correct prediction of cellular responses to IGF1 stimulation in Fig 2B. However, it is apparent that the model consistently over-estimates the effect of small amounts of IGF1 on the downstream pathways. Biological systems frequently operate with thresholds, and possibly the model could be improved, by adding a threshold before which the downstream pathways are not triggered. If this is not feasible, this discrepancy should at least be addressed and explained.

2. Akt activation following IGF1/Ins stimulation

One of the central tenets of the authors is that the two stimulatory pathways differ from each other insofar as S6K inhibition leads to a further increase in AKT phosphorylation downstream of Ins stimulation 30 minutes post exposure, while the same is not true for IGF1 stimulation. They perform a number of validation experiment in a number of cell lines (Fig. 4), which seemingly confirm this idea.

The main problem I see with this set of experiments, is that in all instances the initial AKT phosphorylation following IGF1 stimulation is significantly higher than following Ins stimulation. This makes the effects hard to compare, as pAKT might already have reached maximum phosphorylation levels after 10 minutes of IGF1 stimulation combined with S6K inhibition, with no chance of rising even higher within the following 20 minutes of incubation. Alternatively, completely independent feedback mechanisms might be at play, as biological systems tend to be self-regulatory. In fact, in the case of MCF7 cells, where the difference between the initial levels of pAKT elicited by IGF1 or Ins is not quite as big as in the other cell lines, a clear trend towards a further increase of pAKT after 30 minutes of IGF1 stimulation with S6K inhibition can be seen. Personally, I struggle to believe that significance has not been reached, when comparing control cells to inhibitor treated ones at 30 minutes (top right panel).

I thus propose to repeat at the experiments in at least one cell line, while titrating IGF1 concertation such that it gives an equivalent AKT phosphorylation as Ins stimulation.

**Have the authors made all data and (if applicable) computational code underlying the findings in their manuscript fully available?**

Reviewer #1: Yes

Reviewer #2: Yes

PLOS authors have the option to publish the peer review history of their article (what does this mean?). If published, this will include your full peer review and any attached files.

Reviewer #1: No

Reviewer #2: **Yes: **Peter Kreuzaler
---

## [Decision Letter · Decision Letter 1]

28 May 2021

Dear Dr. Lezon,

We are pleased to inform you that your manuscript 'Inhibition of RPS6K reveals context-dependent Akt activity in luminal breast cancer cells' has been provisionally accepted for publication in PLOS Computational Biology.

Best regards,

Benjamin Hall, DPhil

Guest Editor

PLOS Computational Biology

Florian Markowetz

Deputy Editor

PLOS Computational Biology

Reviewer's Responses to Questions

**Comments to the Authors:**

Reviewer #1: The authors have addressed my comments.

Reviewer #2: I have read the revised version of the manuscript “Inhibition of RPS6K reveals context-dependent Akt activity in luminal breast cancer cells”, by Erdem et al. I thank the author’s for having taken the reviewers’ suggestions into account.

I had originally suggested to perform a few additional cell based experiments. While the authors did not do that, they provided some additional data, as well as addressing all of the points raised in the text, which largely addresses my concerns. They now clearly highlight some of the discrepancies as well as challenges that computational modelling harbours. This shows the reader the quality, but also the shortcomings of any modelling approach. The language is now appropriate to the findings and the claims are backed by the data, which is very good. On the whole I believe this to be a relevant contribution to the field and as such appropriate for publication in PLOS Computational Biology.

**Have the authors made all data and (if applicable) computational code underlying the findings in their manuscript fully available?**

Reviewer #1: Yes

Reviewer #2: Yes

PLOS authors have the option to publish the peer review history of their article (what does this mean?). If published, this will include your full peer review and any attached files.

Reviewer #1: No

Reviewer #2: **Yes: **Peter Kreuzaler

---

## [Editor Report · Acceptance letter]

25 Jun 2021

PCOMPBIOL-D-21-00080R1 

Inhibition of RPS6K reveals context-dependent Akt activity in luminal breast cancer cells

Dear Dr Lezon,

I am pleased to inform you that your manuscript has been formally accepted for publication in PLOS Computational Biology. Your manuscript is now with our production department and you will be notified of the publication date in due course.

With kind regards,

Katalin Szabo
